# Edge Grasp Network: A Graph-Based SE(3)-invariant Approach to Grasp Detection

Haojie Huang    Dian Wang    Xupeng Zhu    Robin Walters    Robert Platt

Khoury College of Computer Science, Northeastern University

{huang.haoj; wang.dian; zhu.xup; r.walters; r.platt} @northeastern.edu

*Abstract*—**Given point cloud input, the problem of 6-DoF grasp pose detection is to identify a set of hand poses in $\mathrm{SE}(3)$ from which an object can be successfully grasped. This important problem has many practical applications. Here we propose a novel method and neural network model that enables better grasp success rates relative to what is available in the literature. The method takes standard point cloud data as input and works well with single-view point clouds observed from arbitrary viewing directions. Videos and code are available at https://haojhuang.github.io/edge_grasp_page/.**

## I. INTRODUCTION

Grasp detection [6, 25, 18] is a critical robotic skill. The robot first observes a scene containing objects in the form of images, voxels, or point clouds, and detects a set of viable grasp poses from which an object may be grasped stably. There are two general approaches: $\mathrm{SE}(2)$ methods where the model reasons in terms of a top-down image of the scene (e.g. [13, 15, 17, 12, 30]), and $\mathrm{SE}(3)$ methods where the model reasons in terms of a point cloud or voxel grid (e.g. [6, 18, 8, 3]). $\mathrm{SE}(3)$ methods have a distinct advantage over $\mathrm{SE}(2)$ methods because they have more flexibility and are easier to apply in general robotics settings. Unfortunately, $\mathrm{SE}(3)$ methods are generally much more complex, so $\mathrm{SE}(2)$ models are often preferred.

This paper tackles the problem of $\mathrm{SE}(3)$ grasping with a novel grasp detection model that we call the *Edge Grasp Network*. The model is based on a novel representation of a 6-DoF grasp that uses a pair of vertices in a graph. Given a single approach point (a position the hand will approach), we define a KNN graph that contains all the points in the point cloud that are within a fixed radius of the approach point. Each point in this KNN graph corresponds to an orientation of the gripper and, when paired with the approach point, defines a distinct 6-DOF grasp pose. We infer the quality of all such grasps simultaneously using a graph neural network.

This approach is novel relative to the literature in three ways: 1) First, our method of defining unique grasp candidates in terms of a pair of vertices in a graph is new; 2) Second, our inference model using a graph neural network defined with respect to a single approach point is novel; 3) Third, our model is the first $\mathrm{SE}(3)$ grasp method that incorporates $\mathrm{SO}(3)$ equivariance.

## II. PROBLEM STATEMENT

The grasp detection problem is to locate a set of grasp poses in $\mathrm{SE}(3)$ for a parallel-jaw gripper given input about the scene in the form of a point cloud. Denote the point cloud observation as $P = \{p_i \in \mathbb{R}^3\}_{i=1}^n$, where $n$ is the number of points. For each point $p \in P$, we will assume that an estimate of the object surface normal $n_p \in S^2$ can be calculated. Although it is not required, we generally assume that this point cloud is generated by a single depth camera. A grasp pose of the gripper is parameterized $\alpha = (C, R) \in \mathrm{SE}(3)$, where $C \in \mathbb{R}^3$ is the location of the center of the gripper and $R \in \mathrm{SO}(3)$ represents its orientation. The grasp detection problem is to find a function $S: P \mapsto \{\alpha_i \in \mathrm{SE}(3)\}_{i=1}^m$, that maps $P$ onto $m$ grasp poses detected in the scene. The grasp evaluation problem is to find a function $\Phi : (P, \alpha) \mapsto [0, 1]$, that denotes the quality of grasp $\alpha$. Notice that $\Phi$ is invariant to translation and rotation in the sense that $\Phi(g \cdot P, g \cdot \alpha) = \Phi(P, \alpha)$ for an arbitrary $g \in \mathrm{SE}(3)$. In other words, the predicted quality of a grasp attempt should be invariant to transformation of the object to be grasped and the grasp pose by the same rotation and translation.

## III. METHOD

### A. Grasp Pose Representation

We represent a grasp as a pair of points in the cloud, $(p_a, p_c) \in P^2$. $p_a$ is considered to be the *approach* point and $p_c$ is the *contact* point. Assuming that we can estimate the object surface normal $n_c$ at point $p_c$, $(p_a, p_c)$ defines a grasp orientation $R$ where the gripper fingers move parallel to the vector $n_c$ and the gripper approaches the object along the vector $a_{ac} =$

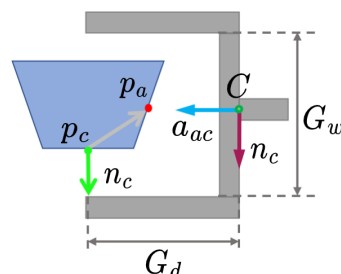

Fig. 1. Grasp pose defined by the edge grasp $(p_a, p_c)$. The reference frame of the gripper is illustrated by the RGB coordinate system. $G_w$ and $G_d$ are the gripper width and gripper depth.

$n_c \times (n_c \times (p_a - p_c))$. This is illustrated in Figure 1. The gripper center $C$ is positioned such that $p_a$ is directly between the fingers and $p_c$ is at a desired point of contact on the finger, $C = p_a - \delta a_{ac}$. Here, $\delta = G_d + (p_a - p_c)^T a_{ac}$ denotes the distance between the center of the gripper and $p_a$ and $G_d$ denotes gripper depth. We will sometimes refer to a grasp defined this way as an *edge grasp*.

To sample edge grasps, we will generally sample the approach point $p_a$ first and then for each approach point sample multiple contact points $p_c$ from the neighbors of $p_a$ within the

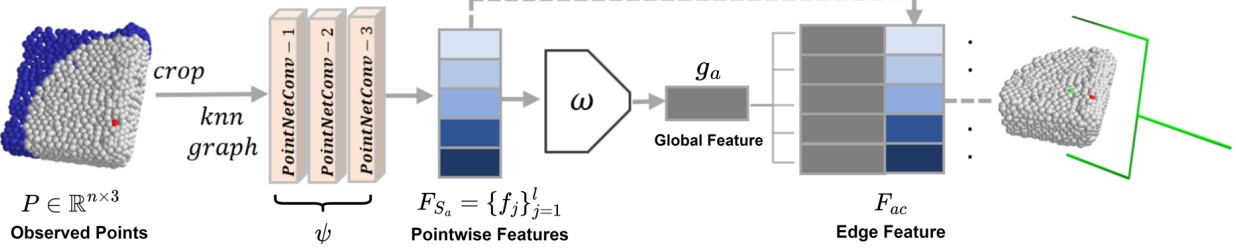

Fig. 2. Encoding process of edge grasps. The rightmost part shows the represented grasp of one edge feature.

distance of $\frac{G_w}{2}$, where $G_w$ denotes the aperture of the gripper, i.e. the distance between the fingers when the gripper is open. One key advantage of this representation is that we can easily provide the approximate position of a desired grasp as an *input* to the model. If we want to grasp a tool by its handle, for example, this is easily achieved by only considering contact locations on the handle.

### B. Model Architecture

Our model, which we call the *Edge Grasp Network*, evaluates the grasp quality for a set of edge grasps that have a single approach point $p_a \in P$ in common. We evaluate multiple approach points by cropping them separately and then placing them in a batch. There are four steps, as illustrated in Figure 2.

Step 1: Crop Point Cloud. Given a point cloud $P$ and an approach point $p_a$, only a set of neighboring points of $p_a$ affects the edge grasp. We crop the point cloud to a ball around $p_a$:

$$S_a = \{p \in P : \|p - p_a\|_2 \leq G_w/2\},$$

Step 2: PointNetConv ($\psi$). We compute a feature at each point using a stack of PointNetConv layers [21], denoted $\psi$. Each layer calculates a new feature $f_i^{(l+1)}$ at each point $p_i \in S_a$ using

$$f_i^{(\ell+1)} = \max_{j \in \mathcal{N}(i)} \mathrm{MLP}\left(f_j^{(\ell)}, p_j - p_i\right), \qquad (1)$$

where $\mathcal{N}(i)$ denotes the $k$-nearest neighbors to $p_i$. Here, $f_j^{(l)}$ denotes the feature at point $p_j$ prior to the layer, $\max$ denotes max-pooling where the max is taken over features (like in PointNet [20]). MLP is a 2-layer multi-layer perceptron that takes both parameters as input. The input features at the first layer are the positions and surface normals of the points. Let $F_{S_a}$ denote the set of features for the points in $S_a$ at the output of Step 2.

Step 3: Compute Global Feature ($\omega$). $\omega$ takes $F_{S_a}$ as input and generates a single global feature $g_a$ that describes $S_a$. First, $F_{S_a}$ is passed to an MLP followed by a max-pooling layer (over features) to generate a first-level global feature. This is concatenated with each feature $f \in F_{S_a}$ and passed to a second MLP and max-pooling layer to output $g_a$. Finally, for each edge grasp $(p_a, p_c) \in P^2$ associated with $p_a$, we calculate an edge feature $f_{ac} \in F_{ac}$ by concatenating $g_a$ with the point feature $f_c \in F_{S_a}$ corresponding to $p_c$. This edge feature will represent the edge grasp to the classifier.

Step 4: Grasp Classification. After calculating the edge features $F_{ac}$, we predict grasp success using a four-layer MLP with a sigmoid function which takes an edge feature $f_{ac}$ as input and infers whether the corresponding edge grasp will succeed.

### C. SO(3) Invariance of Edge Grasp Network

In Section II, we noted that the grasp quality function $\Phi(P, \alpha)$ is invariant to translation and rotation, i.e. $\Phi(g \cdot P, g \cdot \alpha) = \Phi(P, \alpha)$ for arbitrary $g \in \mathrm{SE}(3)$. As presented above, the Edge Grasp Network is invariant to translation because each $S_a$ is centered at the approach point $p_a$ (we translate $p_a$ to the origin of the world frame). However, additional methodology is required to create invariance to rotations. Rotational invariance allows the model to generalize grasp knowledge from one orientation to another. We enable rotational invariance with two different approaches. The first approach is to apply data augmentation on $S_a$ to learn $\mathrm{SO}(3)$ invariance during training. Our second approach is to use an $\mathrm{SO}(3)$-equivariant model, Vector Neurons [5]. Vector Neurons can be applied to nearly any neural model architecture by encoding the $\mathbb{R}^3$ along which $\mathrm{SO}(3)$ acts as a separate tensor axis. As we show in Section IV-C, leveraging $\mathrm{SO}(3)$ symmetries is beneficial to learn a grasp function.

## IV. SIMULATIONS

We benchmarked our method in simulation against three strong baselines, PointNetGPD [14], VGN [2], and GIGA [8]. To make the comparison as fair as possible, we used the same simulator developed by Breyer et al. [2] and used by Jiang et al. [8]. There are two types of simulated grasp environments, PACKED and PILED. In PACKED, objects are placed randomly in an upright configuration in close proximity, e.g. as shown in Figure 3(a). In PILED, objects are dumped randomly from a box into a pile.

### A. Experimental Protocol:

We evaluate our model over several rounds of testing. During each round, a pile or packed scene with 5 test objects is generated inside of a $30 \times 30 \times 30\,\mathrm{cm}^3$ workspace and the system begins grasping one object at a time. Prior to each grasp, we take a depth image of the scene from a direction above the table to extract the point cloud or TSDF, and pass it to the model. After receiving grasp scores from the model, we execute the grasp with the highest quality score. A round of testing ends when either all objects are cleared or two

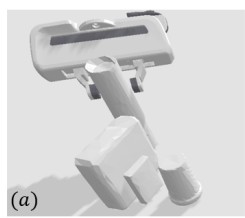 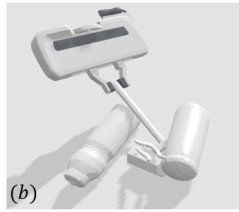

(a)      (b)

Fig. 3. Left: the packed scenario; Right: the pile scenario.

consecutive grasp failures occur. Performance is measured over 100 simulation rounds with 5 different random seeds in terms of: 1) Grasp Success Rate (GSR = $\frac{\#\text{successful grasps}}{\#\text{total grasps}}$); and 2) Declutter Rate (DR = $\frac{\#\text{grasped objects}}{\#\text{total objects}}$). The results are reported in Table I. Detailed description of the baselines and training could be found in Appendix VIII-G and VIII-F.

TABLE I. Quantitative results of clutter removal. *Edge-sample* randomly sample edges that do not collide with the table. *EdgeGraspNet* is the version of our method trained with data augmentation. *VN-EdgeGraspNet* is the version with Vector Neurons. GIGA-High query at a higher resolution of $60 \times 60 \times 60$.

| Method | Packed | | Pile | |
|---|---|---|---|---|
| | GSR (%) | DR (%) | GSR (%) | DR (%) |
| PointNetGPD | $79.3 \pm 1.8$ | $82.5 \pm 2.9$ | $75.6 \pm 2.3$ | $77.0 \pm 2.8$ |
| VGN | $80.2 \pm 1.6$ | $86.2 \pm 2.0$ | $64.9 \pm 2.2$ | $69.1 \pm 3.2$ |
| GIGA | $85.3 \pm 1.9$ | $91.2 \pm 1.7$ | $69.9 \pm 1.8$ | $75.2 \pm 2.2$ |
| GIGA-High | $88.5 \pm 2.0$ | $93.9 \pm 1.4$ | $74.1 \pm 1.5$ | $80.1 \pm 0.5$ |
| Edge-Sample | $44.0 \pm 4.0$ | $39.7 \pm 4.5$ | $40.2 \pm 2.5$ | $30.9 \pm 3.2$ |
| EdgeGraspNet | $92.0 \pm 1.4$ | $94.8 \pm 0.8$ | $89.9 \pm 1.8$ | $92.8 \pm 1.6$ |
| VN-EdgeGraspNet | $\mathbf{92.3 \pm 1.2}$ | $\mathbf{95.2 \pm 0.6}$ | $\mathbf{92.3 \pm 1.5}$ | $\mathbf{93.5 \pm 1.8}$ |

| Method | PointNetGPD | VGN | GIGA | GIGA-High | EdgeGraspNet | VN-EdgeGraspNet |
|---|---|---|---|---|---|---|
| # of Parameters | 1.6 M | 0.3 M | 0.6 M | 0.6 M | 3.0 M | 1.7 M |
| Inference time | 382 ms | 10 ms | 21 ms | 50 ms | 28 ms | 89 ms |

TABLE II. Number of parameters and inference time for proposed methods and baselines. Evaluated on one NVIDIA-GeForce RTX 3090.

### B. Results Analysis:

We draw several conclusions from Table I. First, our sample strategy unadorned with grasp quality inference (Edge-Sample) already performs with a grasp success rate of between 40% and 44%. This suggests our edge grasp representation and sample strategy provide a helpful bias. Second, both Edge-GraspNet and VN-EdgeGraspNet outperform all the baselines in all performance categories by a significant margin, particularly in the PILE category. Third, the performance gap between the packed and piled scenarios is smaller for our method than that for the baselines, which suggests that our model adapts to different object configurations better. Finally, one concern of most sampled-based methods is the inference time since they need to evaluate each grasp individually. However, our method takes use of the shared global features and could achieve a real-time inference time. Detailed inference time analyses could be found in Appendix VIII-H.

### C. Vector Neurons and Data Augmentation:

To investigate the role of $SO(3)$ invariance, we compared our base version of EdgeGraspNet with a variation that omits data augmentation (EdgeGraspNet-NoAug) and VN-EdgeGraspNet.

As shown in Figure 4, the Vector Neurons version performs best and learns fastest, and the base EdgeGrasp-Net converges to approximately the same level. However, without either Vector Neurons or data augmentation, the model overfits. This demonstrates that leveraging $SO(3)$ symmetry is beneficial to learning the grasp function.

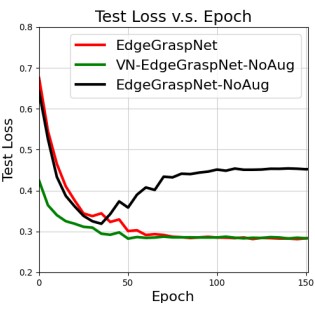

Fig. 4. Test loss functions showing the effect of data augmentation and Vector Neurons.

### D. Ablation study on cropping $S_a$

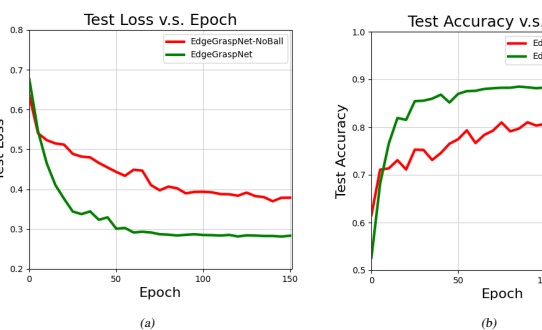

(a)      (b)

Fig. 5. Ablation Study on cropping $S_a$. Left Figure: Test loss v.s. Epoch; Right Figure: Test Accuracy v.s. Epoch. The results show the effect of cropping $S_a$.

We compare our EdgeGrapNet with a variation that skips cropping point cloud around the approach point $p_a$. After getting the observed point cloud $P$, we build a KNN graph on $P$ and feed it to $\psi$ directly to get the point features $F_P$. Then, we extract the global feature $g_a$ corresponding to $p_a$ from $\{f_p \in F_P \mid p \in S_a\}$. Instead of translating $p_a$ to the origin of the world coordinate, we center $P$, the entire observed point cloud, at the origin. Except for these variations, other operations are the same. Let's denote the variation as EdgeGraspNet-NoBall. Figure 5 shows the results of our model and the variation version. It indicates that implementing on $S_a$ is better than implementing on $P$. There are some reasons why $S_a$ is better than $P$. First, $P$ is a special case of $S_a$ when we set the radius of the sphere as infinity. Second, $S_a$ includes all the related points that affect the grasp quality without redundant information. Last but not least, the invariant property on $S_a$ is more generalized than that on $P_a$. Given a $g \in SO(3)$, a grasp action $\alpha$, and a grasp evaluation function $\Psi$, the invariance of EdgeGraspNet could be defined as

$$\Psi(g \cdot S_a, g \cdot \alpha) = \Psi(S_a, \alpha)$$

However, EdgeGraspNet-NoBall could only be invariant to rotations on the entire point cloud: $\Psi(g \cdot P, g \cdot \alpha) = \Psi(P, \alpha)$, which is less generalized.

## V. EVALUATION ON A ROBOT

In this paper, we measure physical grasp performance in three different setups with 4 object sets, as shown in Figure 7. Our model trained in simulation is directly implemented on a real robot.

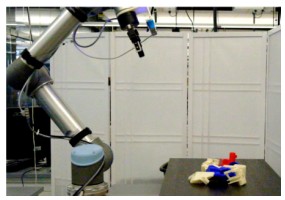 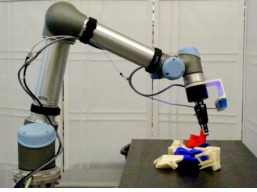

*(a)* *(b)*

Fig. 6. Robot setup. Left: the robot takes a depth image of the scene from a random viewpoint. Right: the robot grasps the red adversarial object from a localized graspable part.

## VI. SETUP

We used a UR5 robot equipped with a Robotiq-85 Gripper, as shown in Figure 6. An Occipital Structure Sensor was mounted on the arm to capture the observation. Prior to each grasp, we move the sensor to a randomly selected viewpoint[1] (pointing toward the objects to be grasped, as shown in Figure 6(a)), take a depth image, and generate a point cloud. We detect and remove the table plane with RANSAC and we denoise and downsample the point cloud using Open3D [29]. For each observed point cloud, we sample 40 approach points and 2000 grasps total. After running inference, we filter out the grasps with a grasp quality score below 0.9. As is the procedure in [2] and [6], we select the highest (largest $z$-coordinate) above-threshold candidate for execution. We use *MoveIt 2* to plan the motion of the robot arm. A grasp is labeled as a success only when the object(s) is picked and transferred to the bin.

### A. Results

Household Objects in the Packed and Pile Settings: This experiment evaluates our method in the *packed* and *piled* settings described in Section IV. In each round, 5 objects are randomly selected from 10 objects. Table III reports grasp success rates and declutter rates from 16 rounds (80 objects total). GSRs vary between 91.7% and 93% – a result that closely matches our simulated results. It indicates the small sim-to-real gap of our method.

| Method | Packed | | Pile | |
|---|---|---|---|---|
| | GSR (%) | DR (%) | GSR (%) | DR (%) |
| EdgeGrasoNet | 91.9 (80/87) | 100 (80/80) | 93.0 (80/86) | 100 (80/80) |
| VN-EdgeGraspNet | 91.7 (78/85) | 98.7 (79/80) | 92.9 (79/85) | 98.7 (79/80) |

TABLE III. Results of real-robot experiments for packed and piled grasp settings.

Comparison with *Zhu et al.* [31] on test hard Objects: This experiment compares our method against the method of Zhu et al. [31], a strong baseline from the literature. In each round, 10 objects are randomly selected and dumped on the table. Table IV shows the results from 15 runs. VN-EdgeGraspNet outperforms [31] by about four percentage points both in terms of the grasp success rate and the declutter rate – a significant improvement against a strong baseline.

[1]We randomly select a viewpoint and repeatedly use it.

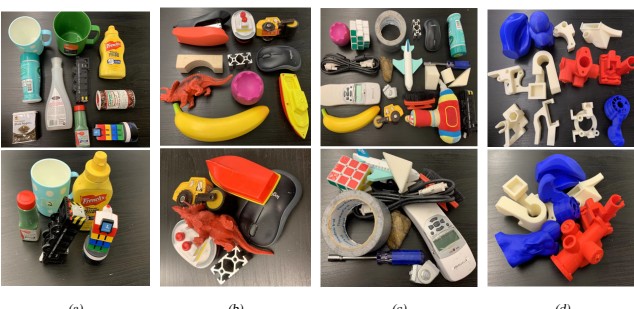

*(a)* *(b)* *(c)* *(d)*

Fig. 7. Object sets and test configurations used for real robot experiments. From left column to right column: packed scene with 10 objects; pile scene with 10 objects; 20 test hard objects [31]; 12 Berkeley adversarial objects [16].

| Method | GSR (%) | DR (%) |
|---|---|---|
| *Zhu et al.* [31] | 89.0 (138/155) | 94.0 (141/150) |
| EdgeGraspNet | 91.8 (146/159) | 98.0 (147/150) |
| VN-EdgeGraspNet | 93.6 (148/159) | 98.6 (148/150) |

TABLE IV. Comparison with the method of Zhu et al. [31] using exactly the same objects and setup.

Comparison with [3] on the Berkeley Adversarial Pile: We also baselined our method using the 12 Berkeley Adversarial Objects described in [16], shown in Figure 7. Here, we compare our method to the work of Cai et al. [3], called *Volumetric Point Network* (VPN). Table V shows the performance comparison. The results indicate that our method outperforms all the baselines. Our final grasp success rate is 84.4%, a very good performance for the Berkeley adversarial object set.

| Method | GSR (%) | DR (%) |
|---|---|---|
| *Gualtieri et al.* [6]* | 70.91 (39/55) | 97.5 (39/40) |
| *Breyer et al.* [2]* | 41.56 (32/77) | 80 (32/40) |
| *Cai et al.* [3]* | 78.4 (40/51) | 100 (40/40) |
| EdgeGraspNet | 84.4 (38/45) | 95.0 (38/40) |
| VN-EdgeGraspNet | 83.0 (40/48) | 100 (40/40) |

TABLE V. Comparison with VPN [3], GPD [6], and VGN [2] for the Berkeley Adversarial Objects in a pile setting. We performed five rounds of grasping with piles of eight objects in each. * Results for VPN [3], GPD [6], and VGN [2] are copied directly from [3].

## VII. CONCLUSION

This paper proposes a novel edge representation in the 6-DoF grasp detection problem. By formulating the grasp pose with an approach point, a contact point, and its surface normal, we represent edge grasps by local features of contacts and global features of the related points. We explore the $SE(3)$ symmetry of our representation and propose EdgeGraspNet and VN-EdgeGraspNet to leverage $SE(3)$ invariance in two different ways. Finally, We evaluate our models on various simulated and real-world object sets against several strong baselines. Experiments show the small sim-to-real gap, the high grasping success rate, and the generalization ability to different object sets of our method. A clear direction for future work is to integrate more on-policy learning, which we believe would enable us to improve our performance.

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

## VIII. APPENDIX

### A. Grasp Sampling

Edge Grasp Network enables us to evaluate a large number of edge grasps that share a single approach point with a single forward pass through the model. However, each different approach point necessitates evaluating the model separately. Therefore we adopt the following grasp sample strategy. First, we sample a small number of approach points $P_a \subset P$. These approach points can be sampled uniformly at random from the cloud, or they can be focused on parts of the cloud where a grasp is preferred. Then, we evaluate the model once for all approach points by forming a minibatch of $|P_a|$ inputs and performing a single forward pass. The output of this is a set of sets of edge grasp features, $F_{(ac)_1}, F_{(ac)_2}, \ldots, F_{(ac)_{|P_a|}}$. One can take the union of these sets, sample $m$ edge grasps uniformly at random or select grasps with preferred gripper approach directions and gripper contact locations, and then run the grasp classifier on these sampled grasps to produce the final output.

### B. Simulator Setting and Grasp Label

Here, we provide a more detailed description of our simulator settings. To generate the training data, we selected a random number of objects from training object sets. We set the mass of each object as $0.5$ kg and the friction ratio between the gripper and the object as $0.75$. We label up to 2000 edge grasp candidates per scene by attempting grasps in simulation. To sample 2000 grasps, we sample 32 approach points from the observed point clouds through Farthest Point Sampling. Edge grasps whose minimum $z$ value (the height) is smaller than the height of the table are filtered out to avoid colliding with the table. A *True* label of a grasp candidate must satisfy the following conditions: 1) the gripper should not collide with any objects when moving from the "prergrasp" pose to the grasp pose; 2) the object must be held by the gripper after a sequence of gripper shaking motions.

### C. Model

We implemented the Edge Grasp Network model described in Section III-B. The input to the model is a downsampled point cloud created by voxelizing the input with a 4mm voxel dimension. The PointNetConv layers in $\psi$ are implemented using a KNN graph with $k = 16$, i.e. with 16 nearest neighbors. $\psi$ is implemented as a sequence of three PointNetConv layers with a 2-layer MLP as the message passing function. The grasp classifier is implemented as a 4-layer MLP with ReLUs [19] and a sigmoid layer at the end. We evaluate both conventional and Vector Neuron versions of our model in simulated and real-robot experiments.

### D. Data Augmentation

Extensive data augmentation is applied to the conventional version of our model to force it to learn the SO(3) invariance from training. Before loading the point cloud $P$ from the training dataset, we randomly sample a $g \in$ SO(3) to rotate $P$. This results in rotations on the 32 cropped point clouds corresponding to each approach point, i.e., $\{g \cdot S_{a_1}, g \cdot S_{a_2}, \ldots, g \cdot S_{a_{32}}\}$.

Since $S_a$ is centered at $p_a$, we then translate $p_a$ to the origin. A batch of 32 rotated and translated $S_a$ is fed to our model as the input during training. Since the Vector Neurons version of our model obtains SO(3) invariance by mathematical constraint, in this case only a translation is applied to each $S_a$.

### E. SO(3) Equivariance to SO(3) Invariance

Based on Vector Neurons [5], we implement the equivairant PointNetConv to realize the SO(3) equivariant feature. We maintain the equivariance of our network until getting the edge feature $f_{ac}$. Invariance is a speical case of equivariance and can be achieved by multiplying a matrix $T_{ac} \in \mathbb{R}^{3 \times 3}$ generated from $f_{ac}$ by a network:

$$(f_{ac}R)(T_{ac}R)^\top = f_{ac}RR^\top T_{ac}^\top = f_{ac}T_{ac}^\top \qquad (2)$$

Equation 2 transforms the SO(3)-equivairant edge feature to SO(3)-invariant edge feature. Combined with the translational invariance described in Section IV-D, we finally realize the SE(3) invariance of edge features. Once the edge features are SE(3) invariant, the entire network becomes SE(3) invariant, i.e., the invariant feature could be fed to a conventional MLP without breaking its invariant property.

### F. Training

The grasp simulator developed by Breyer et al. [2] includes a Franka-Emika Panda gripper. There are 303 training objects and 40 test objects drawn collectively from YCB [4], Big-Bird [24] and other sources [10, 9]. We created training data by generating both packed and piled scenes with a random number of objects in simulation, we add pixelwise Gaussian noise ($\mathcal{N} \sim (0, 0.001)$) to the depth image, extract the point cloud or TSDF (Truncated Signed Distance Function) from the depth image, voxelizing the point cloud with 4-millimeter voxel, generating up to 2000 edge grasp candidates per scene, and labeling each of those candidates by attempting a grasp in simulation. To generate the 2000 edge grasp candidates, we sample 32 approach points uniformly at random from the voxelized cloud. In total, we generated 3.36M labeled grasps based on $3,317$ scenes, 85% of which were used for training and 15% were used for testing. We train our model with the Adam [11] optimizer and an initial learning rate of $10^{-4}$. The learning rate is reduced by a factor of 2 when the test loss has stopped improving for 6 epochs. It takes about 0.5 seconds to complete one SGD step with a batch size of 32 on a NVIDIA Tesla V100 SXM2 GPU. We train the model for 150 epochs and balance the positive and negative grasp labels during training. Both VN-EdgeGraspNet and EdgeGraspNet converge in less than 10 hours.

### G. Baselines for Simulation Experiments

We compare our method against three strong baselines in Section IV. *PointNetGPD* [14] is a sample-based method that represents a candidate grasp pose by the canonicalized points inside the gripper and infers grasp quality using a PointNet [20] model. *VGN* [2] (Volumetric Grasping Network) takes a TSDF of the workspace as input and outputs the grasp orientation and quality at each voxel. *GIGA* [8]

(Grasp detection via Implicit Geometry and Affordance) uses a structured implicit neural representation from 2D feature grids and generates the grasp orientation and quality for each point trained with a auxiliary occupancy loss. Both VGN and GIGA receive a $40 \times 40 \times 40$ TSDF based on output from a single depth image. We also evaluate a variation of GIGA with a $60 \times 60 \times 60$ resolution TSDF, which we refer to as *GIGA-High*. We use the pretrained models[2] of VGN and GIGA from *Jiang et al.* [8] and uniformly sample 64 approach points and 4000 grasps for our method and *PointNetGPD*. As shown in Table II, the pretrained VGN and GIGA models have fewer parameters than our method due to their TSDF input. While our model requires more parameters to operate on point clouds, all compared models are relatively lightweight.

## H. Performance Considerations

Inference Time: Table II shows the time needed by various models to infer grasp qualities. At 28ms per 4,000 grasps, our EdgeGraspNet model is slightly slower than both VGN and GIGA but still much faster than PointNetGPD and GIGA-High. The Vector Neurons version of out model is about three times slower than the EdgeGraspNet model.

| Method | Packed | | Pile | |
|---|---|---|---|---|
| | GSR (%) | DR (%) | GSR (%) | DR (%) |
| EdgeGraspNet (16-1k) | $88.5 \pm 1.7$ | $92.6 \pm 1.4$ | $84.8 \pm 2.1$ | $86.7 \pm 3.3$ |
| EdgeGraspNet (32-2k) | $91.4 \pm 1.5$ | $94.0 \pm 2.0$ | $89.4 \pm 1.3$ | $91.2 \pm 2.5$ |
| EdgeGraspNet (64-4k) | $92.0 \pm 1.4$ | $94.8 \pm 0.8$ | $89.9 \pm 1.8$ | $92.8 \pm 1.6$ |
| VN-EdgeGraspNet (16-1k) | $89.7 \pm 2.4$ | $92.2 \pm 1.6$ | $87.1 \pm 0.8$ | $88.5 \pm 2.3$ |
| VN-EdgeGraspNet (32-2k) | $91.4 \pm 1.3$ | $93.8 \pm 2.0$ | $89.3 \pm 0.5$ | $92.1 \pm 1.8$ |
| VN-EdgeGraspNet (64-4K) | $92.3 \pm 1.2$ | $95.2 \pm 0.6$ | $92.3 \pm 1.5$ | $93.5 \pm 1.8$ |

TABLE VI. Grasp performance for different numbers of approach points (16, 32, and 64) and grasp samples (1000, 2000, and 4000).

TABLE VII. Inference time v.s. # of approach points. We sample different numbers of approach points (16, 32 and 64) with the same number (2000) of edge grasps. Evaluated on one NVIDIA-GeForce RTX 3090.

| | 16-2k | 32-2k | 64-2k |
|---|---|---|---|
| EdgeGraspNet | 9.6 ms | 15.8 ms | 27.4 ms |

| | 32-500 | 32-1k | 32-2k |
|---|---|---|---|
| EdgeGraspNet | 15.8 ms | 15.7 ms | 15.8 ms |

TABLE VIII. Inference time v.s. # sampled edge grasps. We sample different numbers of edge grasps (500, 1000 and 2000) with the same number (32) of approach points. Evaluated on one NVIDIA-GeForce RTX 3090.

Performance of different sample sizes: The speed and performance of our model is closely tied to the number of approach points (which determines batch size) and the number of classified grasps. Table VI shows that fewer approach points and grasp samples reduce grasp success somewhat, but not by

[2]Our trained models for VGN and GIGA on the dataset described above in Section VIII-F did not perform as well as the pretrained models from *Jiang et al.* [8]. It is probably because they train separate models for the PACKED and PILE scenarios with a larger dataset (4M labeled grasps for each scenario). We used their pretained models to do the evaluations.

a huge amount. As shown in Table VII, when we double the number of approach points, the inference time increases about 1.7 times. As shown in Table IX, when we fix the number of approach points and increase the sampled edge grasps, the inference time almost does not change.

## I. Failure Case Analysis

| Method | EdgeGraspNet | | VN-EdgeGraspNet | |
|---|---|---|---|---|
| | GSR (%) | DR (%) | GSR (%) | DR (%) |
| Household Packed | 91.9 (80/87) | 100 (80/80) | 91.7 (78/85) | 98.7 (79/80) |
| Household Pile | 93.0 (80/86) | 100 (80/80) | 92.9 (79/85) | 98.7 (79/80) |
| Test Hard objects | 91.8 (146/159) | 98.0 (147/150) | 93.6 (148/159) | 98.6 (148/150) |
| Berkeley Adversarial | 84.4 (38/45) | 95.0 (38/40) | 83.0 (40/48) | 100 (40/40) |

TABLE IX. Summary of real Robot experiments. We report grasp success rates (GSR) and declutter rates (DR).

We summarized the results of the real-robot experiments in Table IX. Almost half of our failures are caused by colliding with other objects when executing the grasp. It could be mitigated by considering collision when selecting grasps. However, there are some other cases we think readers might want to notice. 1). Occlusion due to partial observation, e.g., a single camera view could only capture a plane of a complex object. 2). Sensor noise. Our model is robust to small noises and leverage the bilateral symmetry of a parallel jaw gripper, i.e., a flip of the calculated surface normal[3] results in a 180° rotation of the gripper along the approach direction. However, if the observation is largely distorted, the proposed edge grasp could be inaccurate since our sampling strategy is closely related to the observed points. There is a trade-off between the precise grasping and the robust grasping. 3). Grasp label of training data. Our binary label of the training data is described in Section VIII-B, but it does not prohibit *true dangerous* grasps. A dangerous grasp could be defined as there is a large change of the pose of the target object when being grasped regardless a successful outcome or not. We believe the true dangerous grasp could cause false-positive predictions when the observation is noisy. Last but no least, failures are the stepping stones to better algorithms in robotics.

## J. Visualization of Grasps

We shows grasp candidates found using our algorithm in Figure 8. The first two rows show three examples of randomly sampled grasp poses for each observed object. The diversity of grasp poses demonstrates our model can provides a high coverage of possible stable grasps. The last row of Figure 8 shows five grasps that share the same contact point. It indicates our model is beneficial to grasping tasks involved with specific contact locations.

## IX. RELATED WORK

### A. 6-DoF gasping methods

There are two main types of 6-DoF grasping methods in recent research. **Sample-based methods** like GPD [25],

[3]A flip of the calculated surfaced normal happens frequently.

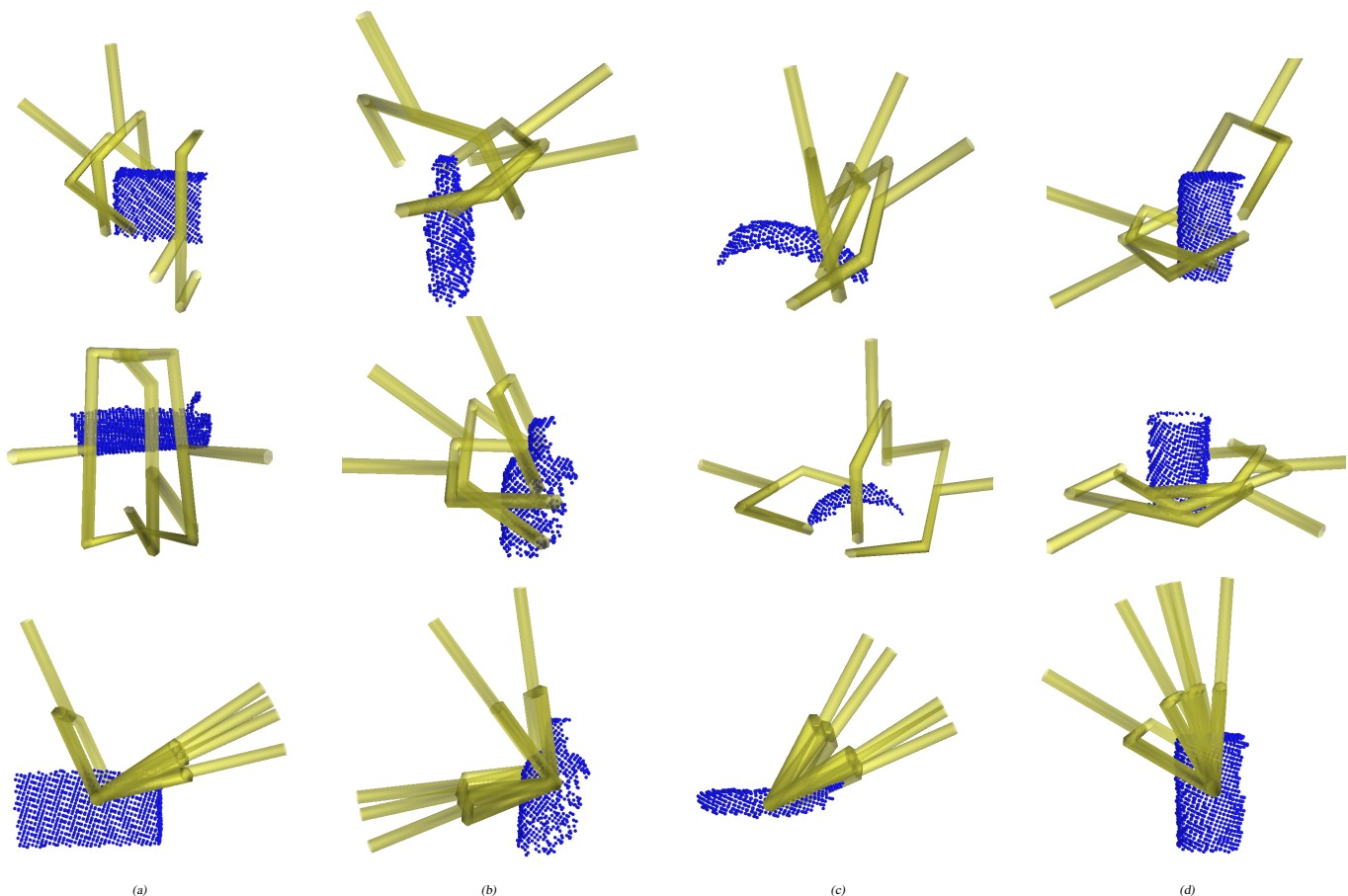

Fig. 8. Illustrations of grasp candidates found using our algorithm. The first two rows show three examples of a gripper placed at randomly sampled grasp candidate configurations. The last row shows five grasps that share the same contact point.

PoinetNetGDP [14], GraspNet [18] that are often comprised of a grasp sampler module and a grasp evaluator module. These methods often require long training time and execution time since each grasp is represented and evaluated individually. In contrast, our method uses shared features to represent different grasps and achieve more computation efficiency. **Element-wise prediction methods** include point-based methods [3, 22, 27, 28] and volumetric-based methods [2, 8]. They estimate grasp qualities for all interesting points or voxels with a single feed-forward propagation. For instance, S4G [22] generates each point feature through PointNet++ [21] and predicts the grasp quality and the grasp pose together. REGNet [28] considers the geometry of radius sphere around the sampled points and regresses the orientations. However, the grasp distribution is a multi-modal function and regression methods only predict one grasp pose for a single point, which may cause ambiguity when multiple graspable poses are valid in that position. Classification methods can generate the distributions over multiple grasps at a single point, but copious amounts of data are often required. Volumetric-based methods [2, 8] use well-structured voxels instead of an unordered set of points. The memory requirements for voxel grids or SDFs are cubic in the resolution of the grid and therefore severely limit the resolution at which the method can be applied.

### B. Grasp Pose Representation

Grasp representation matters in evaluating and refining grasp poses. Most sample-based methods have a clear representation of grasp pose. GPD [25] projects the points around the gripper into canonical planes; PoinetGPD [14] feeds the points inside the gripper to PointNet; GraspNet [18] represents the grasp pose with a set of points of the gripper. On the other hand, element-wise methods [3, 22, 27, 28, 2, 8] often avoid representing grasp explicitly. Since the relative pose between the gripper and the point/voxel is unclear, they have to do regressions or classifications of some elements of the grasp pose. Our method has a clear representation of the grasp pose and satisfies the multi-modal property of the grasp distribution and the friction constraint [1] of the contact point.

### C. Symmetries in Manipulation

Symmetries and equivariance have been shown to improve learning efficiency and generalization ability in many manipulation tasks [31, 26, 7, 23]. *Zhu et al.* [31] decouples rotation and translation symmetries to enable the robot to learn a planar grasp policy within 1.5 hours; *Huang et al.* [7] achieve better sample efficiency and faster convergence speed in planar pick and place tasks with the use of $C_n \times C_n$ equivariance; *Simeonov et al.* [23] use Vector Neurons to get SE(3)-equivariant object representations so that the model can

manipulate objects in the same category with a few training demonstrations. Our method also leverages $\mathrm{SE}(3)$ symmetry to learn faster and generalize better on 6-DoF grasping.