# OpenReview forum: "Edge Grasp Network: A Graph-Based SE(3)-invariant Approach to Grasp Detection"
_roboticsfoundation.org/RSS/2023/Workshop/Symmetry — RSS 2023 Workshop Symmetry_

### Official Review · Reviewer_dp8W · 2023-06-18
**Review for Edge Grasp Network**

**Rating:** 7
**Confidence:** 4

**Review:**

**Summary**

This work proposed the Edge Grasp Network, a novel graph-based SE(3)-invariant model to detect grasp poses. In particular, it

1. proposed a novel and compact representation of the 6-DoF grasp, termed “edge grasp”. It is defined as a pair of vertices in a graph, one represents the approach point, and another for the contact point;
2. based on the proposed edge grasp representation, the proposed inference model utilized a GNN relative to each approach point to infer the quality of candidate grasp poses;
3. incorporated SO(3) equivariance into the SE(3) grasp method, to enable built-in SO(3) invariance desired by the grasp evaluation function, $\Phi: (P, \alpha) \rightarrow [0,1]$

**Strengths**
1. Well-written and well-organized paper, with clear illustrations of the grasp pose representation formulation (Fig 1) and the encoding process (Fig 2).
2. It provided ablation experiments on design choices of 1) different approaches to achieve SO(3) invariance for edge grasp network (data augmentation vs incorporating Vector Neurons); 2) whether to use the cropped point cloud $S_a$ or the whole point cloud; 3) different sample size for approach points.
3. The authors proposed a novel representation for defining grasp candidates, via a pair of vertices in a graph, termed “edge grasp”. By decoupling the grasp pose representation into two points (the approach point and the contact point), this representation can easily incorporate prior shape/task-specific knowledge for grasping, by providing a coarse region of highly probable approach point region, and then learning to predict the associated contact point, which could potentially further improve its sample-efficiency.
4. The authors reported extensive experimental results both in simulation and on physical robots, with a range of test objects in the real world (20 test hard objects and 12 Berkeley adversarial objects). Furthermore, the proposed method seems to have little sim2real gap.
5. The proposed model is able to achieve fast inference time of 28/89 ms, which is needed for real-time control.
6. The authors have provided open-sourced codebase which will be valuable for the community to reproduce and build upon this work.

Overall, this paper is well-written with clear illustrations and notations, with adequate experimental results in both simulation and the real world. The need for SE(3) invariance is adequately motivated. The proposed 6-DoF grasp representation is novel, accompanied by a self-contained model architecture design.

**Weaknesses**
1. *Assumption of availability of surface normal at contact point* - The accuracy and usability of the grasp candidate produced based on the proposed grasp pose representation is a function of the availability and accuracy of the object surface normal $n_c$ at the contact point $p_c$. While it is usually fair to assume relatively accurate surface normal are usually readily obtainable from point cloud data, the accuracy of the normal vector might be susceptible to factors such as **a)** the quality of the captured point cloud (especially in the real world, point cloud captures qualities are an artifact of the depth sensor choices), and **b)** the density of point cloud that’s fed to the model. Potential discussions and analysis on how different depth sensors and point cloud density could affect the surface normal calculation, and thus the performance of the predicted edge pose could be added. Moreover, if the predicted contact point $p_c$ is not on the surface of the object, which will happen at some time during the prediction process after the point cloud is cropped to a ball around $p_a$, more details should be provided about how the surface normal is calculated for points that are not originally on the object surface.
2. Details on **a)** how a successful grasp is evaluated in simulation and the real world, and **b)** motion planning and controller details, are missing. While these are not directly related to the core contribution of the paper, they are important details that should be added in Appendix.

**Comments**

1. It’s interesting that in simulation, VN-EdgeGraspNet outperformed EdgeGraspNet (Table I), but in real-world experiments, with the exception of GSR for Packed, EdgeGraspNet outperforms VN-EdgeGraspNet (Table III). Curious whether the authors have any comments/insights on what might have led to the reverse performance between the simulation and real world, of these two proposed networks.

---

### Official Review · Reviewer_vBq1 · 2023-06-19
**Review of EdgeGraspNet**

**Rating:** 8
**Confidence:** 4

**Review:**

The paper presents the Edge Grasp Network, an approach for detecting successful grasp poses based on point cloud input using a graph-based SE(3)-invariant approach. The method introduces a new grasp pose representation called edge grasp and presents an edge encoding pipeline that calculates local and global point cloud features to compute edge features. These features are then utilized to classify grasp success. The network incorporates SE(3) equivariance. The paper provides a thorough evaluation of the method and baselines in both simulation and real-world scenarios, demonstrating that the Edge Grasp Network outperforms baselines with high grasping success rates.

Pros:
1. The paper presents a novel grasp representation and SE(3) grasp detection method.
2. The paper presents detailed comparisons with existing works and performance analysis. The evaluation of a real robot demonstrates the practical applicability of the method with a small sim-to-real gap, a high grasping success rate, and the generalization ability to different object sets.
3. The presentation of the approach and results is clear.

Cons:
1. The current setting is to remove all objects in a clutter away, and the current pipeline samples approach points randomly. It is unknown if the method can be applied to the setting of grasping specific objects from a clutter since it would require an instance segmentation module and the ability to handle more severe occlusions.
2. The pipeline lacks collision checking, which is necessary for real-world applications.

---

### Decision · Program_Chairs · 2023-06-23

**Decision:**

Accept

**Comment:**

Congratulations! We encourage the authors to revise the paper based on the reviewer's feedback.
Your paper will be presented as both a short presentation and a poster. Detailed instructions about the presentation format and camera-ready submission will be sent to you soon.